# Scalable Lévy Process Priors for Spectral Kernel Learning

**Phillip A. Jang    Andrew E. Loeb    Matthew B. Davidow    Andrew Gordon Wilson**
Cornell University

## Abstract

Gaussian processes are rich distributions over functions, with generalization prop­erties determined by a kernel function. When used for long-range extrapolation, predictions are particularly sensitive to the choice of kernel parameters. It is therefore critical to account for kernel uncertainty in our predictive distributions. We propose a distribution over kernels formed by modelling a spectral mixture density with a Lévy process. The resulting distribution has support for all sta­tionary covariances—including the popular RBF, periodic, and Matérn kernels— combined with inductive biases which enable automatic and data efficient learn­ing, long-range extrapolation, and state of the art predictive performance. The proposed model also presents an approach to spectral regularization, as the Lévy process introduces a sparsity-inducing prior over mixture components, allowing automatic selection over model order and pruning of extraneous components. We exploit the algebraic structure of the proposed process for $\mathcal{O}(n)$ training and $\mathcal{O}(1)$ predictions. We perform extrapolations having reasonable uncertainty estimates on several benchmarks, show that the proposed model can recover flexible ground truth covariances and that it is robust to errors in initialization.

## 1  Introduction

Gaussian processes (GPs) naturally give rise to a *function space* view of modelling, whereby we place a prior distribution over functions, and reason about the properties of likely functions under this prior (Rasmussen & Williams, 2006). Given data, we then infer a posterior distribution over functions to make predictions. The generalisation behavior of the Gaussian process is determined by its prior support (which functions are a priori possible) and its inductive biases (which functions are a priori likely), which are in turn encoded by a kernel function. However, popular kernels, and even multiple kernel learning procedures, typically cannot extract highly expressive hidden representations, as was envisaged for neural networks (MacKay, 1998; Wilson, 2014).

To discover such representations, recent approaches have advocated building more expressive ker­nel functions. For instance, spectral mixture kernels (Wilson & Adams, 2013b) were introduced for flexible kernel learning and extrapolation, by modelling a spectral density with a scale-location mix­ture of Gaussians, with promising results. However, Wilson & Adams (2013b) specify the number of mixture components by hand, and do not characterize uncertainty over the mixture hyperparameters.

As kernel functions become increasingly expressive and parametrized, it becomes natural to also adopt a function space view of kernel learning—to represent uncertainty over the values of the kernel function, and to reflect the belief that the kernel does not have a simple form. Just as we use Gaussian processes over functions to model data, we can apply the function space view a step further in a hierarchical model—with a prior distribution over kernels.

In this paper, we introduce a scalable distribution over kernels by modelling a spectral density, the Fourier transform of a kernel, with a Lévy process. We consider both scale-location mixtures of Gaussians and Laplacians as basis functions for the Lévy process, to induce a prior over kernels that

gives rise to the sharply peaked spectral densities that often occur in practice—providing a powerful inductive bias for kernel learning. Moreover, this choice of basis functions allows our kernel function, conditioned on the Lévy process, to be expressed in closed form. This prior distribution over kernels also has support for *all* stationary covariances—containing, for instance, any composition of the popular RBF, Matérn, rational quadratic, gamma-exponential, or spectral mixture kernels. And unlike the spectral mixture representation in Wilson & Adams (2013b), this proposed process prior allows for natural *automatic* inference over the number of mixture components in the spectral density model. Moreover, the priors implied by popular Lévy processes such as the gamma process and symmetric $\alpha$-stable process result in even stronger complexity penalties than $\ell_1$ regularization, yielding sparse representations and removing mixture components which fit to noise.

Conditioned on this distribution over kernels, we model data with a Gaussian process. To form a predictive distribution, we take a Bayesian model average of GP predictive distributions over a large set of possible kernel functions, represented by the support of our prior over kernels, weighted by the posterior probabilities of each of these kernels. This procedure leads to a non-Gaussian heavy-tailed predictive distribution for modelling data. We develop a reversible jump MCMC (RJ-MCMC) scheme (Green, 1995) to infer the posterior distribution over kernels, including inference over the number of components in the Lévy process expansion. For scalability, we pursue a structured kernel interpolation (Wilson & Nickisch, 2015) approach, in our case exploiting algebraic structure in the Lévy process expansion, for $\mathcal{O}(n)$ inference and $\mathcal{O}(1)$ predictions, compared to the standard $\mathcal{O}(n^3)$ and $\mathcal{O}(n^2)$ computations for inference and predictions with Gaussian processes. Flexible distributions over kernels will be especially valuable on large datasets, which often contain additional structure to learn rich statistical representations.

The key contributions of this paper are summarized as follows:

1. The first fully probabilistic approach to inference with spectral mixture kernels — to incorporate kernel uncertainty into our predictive distributions, for a more realistic coverage of extrapolations. This feature is demonstrated in Section 5.3.
2. Spectral regularization in spectral kernel learning. The Lévy process prior acts as a sparsity-inducing prior on mixture components, automatically pruning extraneous components. This feature allows for automatic inference over model order, a key hyperparameter which must be hand tuned in the original spectral mixture kernel paper.
3. Reduced dependence on a good initialization, a key practical improvement over the original spectral mixture kernel paper.
4. A conceptually natural and interpretable function space view of kernel learning.

## 2 Background

We provide a review of Gaussian and Lévy processes as models for prior distributions over functions.

### 2.1 Gaussian Processes

A stochastic process $f(\mathbf{x})$ is a Gaussian process (GP) if for any finite collection of inputs $X = \{\mathbf{x}_1, \cdots, \mathbf{x}_n\} \subset \mathbb{R}^D$, the vector of function values $[f(\mathbf{x}_1), \cdots, f(\mathbf{x}_n)]^T$ is jointly Gaussian.

The distribution of a GP is completely determined by its mean function $m(\mathbf{x})$, and covariance kernel $k(\mathbf{x}, \mathbf{x}')$. A GP used to specify a distribution over functions is denoted as $f(\mathbf{x}) \sim \mathcal{GP}(m(\mathbf{x}), k(\mathbf{x}, \mathbf{x}'))$, where $\mathbb{E}[f(x_i)] = m(\mathbf{x}_i)$ and $\mathrm{cov}(f(\mathbf{x}), f(\mathbf{x}')) = k(\mathbf{x}, \mathbf{x}')$. The generalization properties of the GP are encoded by the covariance kernel and its hyperparameters.

By exploiting properties of joint Gaussian variables, we can obtain closed form expressions for conditional mean and covariance functions of unobserved function values given observed function values. Given that $f(\mathbf{x})$ is observed at $n$ training inputs $X$ with values $\mathbf{f} = [f(\mathbf{x}_1), \cdots, f(\mathbf{x}_n)]^T$, the predictive distribution of the unobserved function values $\mathbf{f}_*$ at $n_*$ testing inputs $X_*$ is given by

$$\mathbf{f}_* | X_*, X, \theta \sim \mathcal{N}(\bar{\mathbf{f}}_*, \mathrm{cov}(\mathbf{f}_*)), \tag{1}$$

$$\bar{\mathbf{f}}_* = m_{X_*} + K_{X_*,X} K_{X,X}^{-1} (\mathbf{f} - m_X), \tag{2}$$

$$\mathrm{cov}(\mathbf{f}_*) = K_{X_*,X_*} - K_{X_*,X} K_{X,X}^{-1} K_{X,X_*}. \tag{3}$$

where $K_{X_*,X}$ for example denotes the $n_* \times n$ matrix of covariances evaluated at $X_*$ and $X$.

The popular radial basis function (RBF) kernel has the following form:

$$k_{\text{RBF}}(\mathbf{x}, \mathbf{x}') = \exp(-0.5 \left\| \mathbf{x} - \mathbf{x}' \right\|^2 / \ell^2). \tag{4}$$

GPs with RBF kernels are limited in their expressiveness and act primarily as smoothing interpolators, because the only covariance structure they can learn from data is the length scale $\ell$, which determines how quickly covariance decays with distance.

Wilson & Adams (2013b) introduce the more expressive *spectral mixture* (SM) kernel capable of extracting more complex covariance structures than the RBF kernel, formed by placing a scale-location mixture of Gaussians in the spectrum of the covariance kernel. The RBF kernel in comparison can only model a single Gaussian centered at the origin in frequency (spectral) space.

## 2.2 Lévy Processes

A stochastic process $\{L(\omega)\}_{\omega \in \mathbb{R}^+}$ is a *Lévy process* if it has stationary, independent increments and it is continuous in probability. In other words, $L$ must satisfy

1. $L(0) = 0$,
2. $L(\omega_0), L(\omega_1) - L(\omega_0), \cdots, L(\omega_n) - L(\omega_{n-1})$ are independent $\forall \omega_0 \leq \omega_1 \leq \cdots \leq \omega_n$,
3. $L(\omega_2) - L(\omega_1) \stackrel{d}{=} L(\omega_2 - \omega_1) \quad \forall \omega_2 \geq \omega_1$,
4. $\lim_{h \to 0} \mathbb{P}(|L(\omega + h) - L(\omega)| \geq \varepsilon) = 0 \quad \forall \varepsilon > 0 \, \forall \omega \geq 0$.

By the Lévy-Khintchine representation, the distribution of a (pure jump) Lévy process is completely determined by its Lévy measure. That is, the characteristic function of $L(\omega)$ is given by:

$$\log \mathbb{E}[e^{iuL(\omega)}] =$$

$$\omega \int_{\mathbb{R}^d \setminus \{0\}} \left( e^{iu \cdot \beta} - 1 - iu \cdot \beta 1_{|\beta| \leq 1} \right) \nu(d\beta).$$

where the Lévy measure $\nu(d\beta)$ is any $\sigma$-finite measure which satisfies the following integrability condition

$$\int_{\mathbb{R}^d \setminus \{0\}} (1 \wedge \beta^2) \nu(d\beta) < \infty.$$

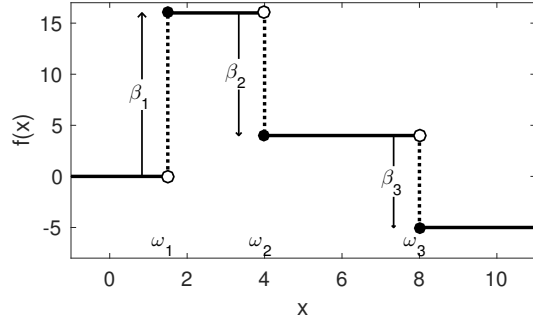

Figure 1: Annotated realization of a compound Poisson process, a special case of a Lévy process. The $\omega_j$ represent jump locations, and $\beta_j$ represent jump magnitudes.

A Lévy process can be viewed as a combination of a Brownian motion with drift and a superposition of independent Poisson processes with differing jump sizes $\beta$. The Lévy measure $\nu(d\beta)$ determines the expected number of Poisson events per unit of time for any particular jump size $\beta$. The Brownian component of a Lévy process will not be considered for this model. For higher dimension input spaces $\omega \in \Omega$, one defines the more general notion of Lévy random measure, which is also characterized by its Lévy measure $\nu(d\beta d\omega)$ (Wolpert et al., 2011) . We will show that the sample realizations of Lévy processes can be used to draw sample parameters for adaptive basis expansions.

## 2.3 Lévy Process Priors over Adaptive Expansions

Suppose we wish to specify a prior over the class of adaptive expansions: $\left\{ f : \mathcal{X} \to \mathbb{R} \mid f(x) = \sum_{j=1}^{J} \beta_j \phi(x, \omega_j) \right\}$. Through a simple manipulation, we can rewrite $f(x)$ into the form of a stochastic integral:

$$f(x) = \sum_{j=1}^{J} \beta_j \phi(x, \omega_j) = \sum_{j=1}^{J} \beta_j \int_{\Omega} \phi(x, \omega) \delta_{\omega_j}(\omega) d\omega = \int_{\Omega} \phi(x, \omega) \underbrace{\sum_{j=1}^{J} \beta_j \delta_{\omega_j}(\omega) d\omega}_{=dL(\omega)}.$$

Hence, by specifying a prior for the measure $L(\omega)$, we can simultaneously specify a prior for all of the parameters $\{J, (\beta_1, \omega_1), ..., (\beta_J, \omega_J)\}$ of the expansion. Lévy random measures provide a

family of priors naturally suited for this purpose, as there is a one-to-one correspondence between the jump behavior of the Lévy prior and the components of the expansion.

To illustrate this point, suppose the basis function parameters $\omega_j$ are one-dimensional and consider the integral of $dL(\omega)$ from 0 to $\omega$.

$$L(\omega) = \int_0^\omega dL(\xi) = \int_0^\omega \sum_{j=1}^J \beta_j \delta_{\omega_j}(\xi) d\xi = \sum_{j=1}^J \beta_j 1_{[0,\omega]}(\omega_j).$$

We see in Figure 1 that $\sum_{j=1}^J \beta_j 1_{[0,\omega]}(\omega_j)$ resembles the sample path of a compound Poisson process, with the number of jumps $J$, jump sizes $\beta_j$, and jump locations $\omega_j$ corresponding to the number of basis functions, basis function weights, and basis function parameters respectively. We can use a compound Poisson process to define a prior over all such piecewise constant paths. More generally, we can use a Lévy process to define a prior for $L(\omega)$.

Through the Lévy-Khintchine representation, the jump behavior of the prior is characterized by a Lévy measure $\nu(d\beta d\omega)$ which controls the mean number of Poisson events in every region of the parameter space, encoding the inductive biases of the model. As the number of parameters in this framework is random, we use a form of trans-dimensional reversible jump Markov chain Monte Carlo (RJ-MCMC) to sample the parameter space (Green, 2003).

Popular Lévy processes such as the gamma process, symmetric gamma process, and the symmetric $\alpha$-stable process each possess desirable properties for different situations. The gamma process is able to produce strictly positive gamma distributed $\beta_j$ without transforming the output space. The symmetric gamma process can produce both positive and negative $\beta_j$, and according to Wolpert et al. (2011) can achieve nearly all the commonly used isotropic geostatistical covariance functions. The symmetric $\alpha$-stable process can produce heavy-tailed distributions for $\beta_j$ and is appropriate when one might expect the basis expansion to be dominated by a few heavily weighted functions.

While one could dispense with Lévy processes and place Gaussian or Laplace priors on $\beta_j$ to obtain $\ell_2$ or $\ell_1$ regularization on the expansions, respectively, a key benefit particular to these Lévy process priors are that the implied priors on the coefficients yield even stronger complexity penalties than $\ell_1$ regularization. This property encourages sparsity in the expansions and permits scalability of our MCMC algorithm. Refer to the supplementary material for an illustration of the joint priors on coefficients, which exhibit concave contours in contrast to the convex elliptical and diamond contours of $\ell_2$ and $\ell_1$ regularization. Furthermore, in the log posterior for the Lévy process there is a $\log(J!)$ complexity penalty term which further encourages sparsity in the expansions. Refer to Clyde & Wolpert (2007) for further details.

## 3   Lévy Distributions over Kernels

In this section, we motivate our choice of prior over kernel functions and describe how to generate samples from this prior distribution in practice.

### 3.1   Lévy Kernel Processes

By Bochner's Theorem (1959), a continuous stationary kernel can be represented as the Fourier dual of a spectral density:

$$k(\tau) = \int_{\mathbb{R}^D} S(s)e^{2\pi i s^\top \tau} ds, \quad S(s) = \int_{\mathbb{R}^D} k(\tau)e^{-2\pi i s^\top \tau} d\tau. \tag{5}$$

Hence, the spectral density entirely characterizes a stationary kernel. Therefore, it can be desirable to model the spectrum rather than the kernel, since we can then view kernel estimation through the lens of density estimation. In order to emulate the sharp peaks that characterize frequency spectra of natural phenomena, we model the spectral density with a location-scale mixture of Laplacian components:

$$\phi_L(s, \omega_j) = \frac{\lambda_j}{2} e^{-\lambda_j |s - \chi_j|}, \quad \omega_j \equiv (\chi_j, \lambda_j) \in [0, f_{max}] \times \mathbb{R}^+. \tag{6}$$

Then the full specification of the symmetric spectral mixture is

$$S(s) = \frac{1}{2}\left[\tilde{S}(s) + \tilde{S}(-s)\right], \quad \tilde{S}(s) = \sum_{j=1}^J \beta_j \phi_L(s, \omega_j). \tag{7}$$

As Laplacian spikes have a closed form inverse Fourier transform, the spectral density $S(s)$ represents the following kernel function:

$$k(\tau) = \sum_{j=1}^{J} \beta_j \frac{\lambda_j^2}{\lambda_j^2 + 4\pi^2\tau^2} \cos(2\pi\chi_j\tau). \tag{8}$$

The parameters $J$, $\beta_j$, $\chi_j$, $\lambda_j$ can be interpreted through Eq. (8). The total number of terms to the mixture is $J$, while $\beta_j$ is the scale of the $j^{\text{th}}$ frequency contribution, $\chi_j$ is its central frequency, and $\lambda_j$ governs how rapidly the term decays (a high $\lambda$ results in confident, long-term periodic extrapolation).

Other basis functions can be used in place of $\phi_L$ to model the spectrum as well. For example, if a Gaussian mixture is chosen, along with maximum likelihood estimation for the learning procedure, then we obtain the spectral mixture kernel (Wilson & Adams, 2013b).

As the spectral density $S(s)$ takes the form of an adaptive expansion, we can define a Lévy prior over all such densities and hence all corresponding kernels of the above form. For a chosen basis function $\phi(s,\omega)$ and Lévy measure $\nu(d\beta d\omega)$ we say that $k(\tau)$ is drawn from a **Lévy kernel process (LKP)**, denoted as $k(\tau) \sim \mathcal{LKP}(\phi,\nu)$. Wolpert et al. (2011) discuss the necessary regularity conditions for $\phi$ and $\nu$. In summary, we propose the following hierarchical model over functions

$$f(x)|k(\tau) \sim \mathcal{GP}(0, k(\tau)), \quad \tau = x - x', \quad k(\tau) \sim \mathcal{LKP}(\phi,\nu). \tag{9}$$

Figure 2 shows three samples from the Lévy process specified through Eq. (7) and their corresponding covariance kernels. We also show one GP realization for each of the kernel functions. By placing a Lévy process prior over spectral densities, we induce a Lévy kernel process prior over stationary covariance functions.

### 3.2 Sampling Lévy Priors

We now discuss how to generate samples from the Lévy kernel process in practice. In short, the kernel parameters are drawn according to $\{J, \{(\beta_j, \omega_j)\}_{j=1}^J\} \sim \text{Lévy}(\nu(d\beta d\omega))$, and then Eq. (8) is used to evaluate $k \sim \mathcal{LKP}(\phi_L,\nu)$ at values of $\tau$.

Recall from Section 2.3 that the choice of Lévy measure $\nu$ is completely determined by the choice of the corresponding Lévy process and vice versa. Though the processes mentioned there produce sample paths with infinitely many jumps (and cannot be sampled directly), almost all jumps are infinitesimally small, and therefore these processes can be approximated in $L^2$ by a compound Poisson process with a jump size distribution truncated by $\varepsilon$.

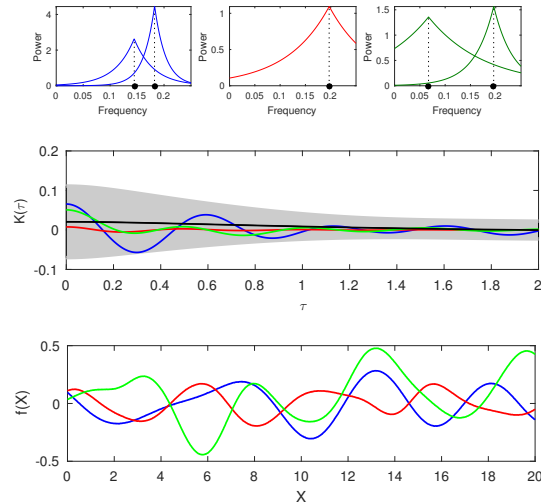

Figure 2: Samples from a Lévy kernel mixture prior distribution. (top) Three spectra with Laplace components drawn from a Lévy process prior. (middle) The corresponding stationary covariance kernel functions and the prior mean with two standard deviations of the model, as determined by 10,000 samples. (bottom) GP samples with the respective covariance kernel functions.

Once the desired Lévy process is chosen and the truncation bound is set, the basis expansion parameters are generated by drawing $J \sim \text{Poisson}(\nu_\varepsilon^+)$, and then drawing $J$ i.i.d. samples $\beta_1, \cdots, \beta_J \sim \pi_\beta(d\beta)$, and $J$ i.i.d. samples $\omega_1, \cdots, \omega_J \sim \pi_\omega(d\omega)$. Refer to the supplementary material for $L^2$ error bounds and formulas for $\nu_\varepsilon^+ = \nu_\varepsilon(\mathbb{R} \times \Omega)$ for the gamma, symmetric gamma, and symmetric $\alpha$-stable processes.

The form of $\pi_\beta(\beta_j)$ also depends on the choice of Lévy process and can be found in the supplementary material, with further details in Wolpert et al. (2011). We choose to draw $\chi$ from an uninformed uniform prior over a reasonable range in the frequency domain, and $\lambda$ from a gamma distribution, $\lambda \sim \text{Gamma}(a_\lambda, b_\lambda)$. The choices for $a_\lambda$, $b_\lambda$, and the frequency limits are left as hyperparameters, which can have their own hyperprior distributions. After drawing the $3J$ values that specify

a Lévy process realization, the corresponding covariance function can be evaluated through the analytical expression for the inverse Fourier transform (e.g. Eq. (8) for Laplacian frequency mixture components).

## 4  Scalable Inference

Given observed data $\mathcal{D} = \{x_i, y_i\}_{i=1}^N$, we wish to infer $p(y(x_*)|D, x_*)$ over some test set of inputs $x_*$ for interpolation and extrapolation. We model observations $y(x)$ with a hierarchical model:

$$y(x)|f(x) = f(x) + \varepsilon(x), \quad \varepsilon(x) \overset{\text{iid}}{\sim} N(0, \sigma^2), \tag{10}$$

$$f(x)|k(\tau) \sim \mathcal{GP}(0, k(\tau)), \quad \tau = x - x', \tag{11}$$

$$k(\tau) \sim \mathcal{LKP}(\phi, \nu). \tag{12}$$

Computing the posterior distributions by marginalizing over the LKP will yield a heavy-tailed non-Gaussian process for $y(x_*) = y_*$ given by an infinite Gaussian mixture model:

$$p(y_*|\mathcal{D}) = \int p(y_*|k, \mathcal{D}) p(k|\mathcal{D}) dk \approx \frac{1}{H} \sum_{h=1}^{H} p(y_*|k_h), \quad k_h \sim p(k|\mathcal{D}). \tag{13}$$

We compute this approximating sum using $H$ RJ-MCMC samples (Green, 2003). Each sample draws a kernel from the posterior $k_h \sim p(k|\mathcal{D})$ distribution. Each sample of $k_h$ enables us to draw a sample from the posterior predictive distribution $p(y_*|\mathcal{D})$, from which we can estimate the predictive mean and variance.

Although we have chosen a Gaussian observation model in Eq. (10) (conditioned on $f(x)$), all of the inference procedures we have introduced here would also apply to non-Gaussian likelihoods, such as for Poisson processes with Gaussian process intensity functions, or classification.

The sum in Eq. (13) requires drawing kernels from the distribution $p(k|\mathcal{D})$. This is a difficult distribution to approximate, particularly because there is not a fixed number of parameters as $J$ varies. We employ RJ-MCMC, which extends the capability of conventional MCMC to allow sequential samples of different dimensions to be drawn (Green, 2003). Thus, a posterior distribution is not limited to coefficients and other parameters of a fixed basis expansion, but can represent a changing number of basis functions, as required by the description of Lévy processes described in the previous section. Indeed, RJ-MCMC can be used to automatically learn the appropriate number of basis functions in an expansion. In the case of spectral kernel learning, inferring the number of basis functions corresponds to automatically learning the important frequency contributions to a GP kernel, which can lead to new interpretable insights into our data.

### 4.1  Initialization Considerations

The choice of an initialization procedure is often an important practical consideration for machine learning tasks due to severe multimodality in a likelihood surface (Neal, 1996). In many cases, however, we find that spectral kernel learning with RJ-MCMC can automatically learn salient frequency contributions with a simple initialization, such as a uniform covering over a broad range of frequencies with many sharp peaks. The frequencies which are not important in describing the data are quickly attenuated or removed within RJ-MCMC learning. Typically only a few hundred RJ-MCMC iterations are needed to discover the salient frequencies in this way.

Wilson (2014) proposes an alternative structured approach to initialization in previous spectral kernel modelling work. First, pass the (squared) data through a Fourier transform to obtain an empirical spectral density, which can be treated as observed. Next, fit the empirical spectral density using a standard Gaussian mixture density estimation procedure, assuming a fixed number of mixture components. Then, use the learned parameters of the Gaussian mixture as an initialization of the spectral mixture kernel hyperparameters, for Gaussian process marginal likelihood optimization. We observe successful adaptation of this procedure to our Lévy process method, replacing the approximation with Laplacian mixture terms and using the result to initialize RJ-MCMC.

### 4.2  Scalability

As with other GP based kernel methods, the computational bottleneck lies in the evaluation of the log marginal likelihood during MCMC, which requires computing $(K_{X,X} + \sigma^2 I)^{-1} y$ and

$\log |K_{X,X} + \sigma^2 I|$ for an $n \times n$ kernel matrix $K_{X,X}$ evaluated at the $n$ training points $X$. A direct approach through computing the Cholesky decomposition of the kernel matrix requires $\mathcal{O}(n^3)$ computations and $\mathcal{O}(n^2)$ storage, restricting the size of training sets to $\mathcal{O}(10^4)$. Furthermore, this computation must be performed at every iteration of RJ-MCMC, compounding standard computational constraints.

However, this bottleneck can be readily overcome through the Structured Kernel Interpolation approach introduced in Wilson & Nickisch (2015), which approximates the kernel matrix as $\tilde{K}_{X,X'} = M_X K_{Z,Z} M_{X'}^\top$, for an exact kernel matrix $K_{Z,Z}$ evaluated on a much smaller set of $m \ll n$ inducing points, and a sparse interpolation matrix $M_X$ which facilitates fast computations. The calculation reduces to $\mathcal{O}(n + g(m))$ computations and $\mathcal{O}(n + g(m))$ storage. As described in Wilson & Nickisch (2015), we can impose Toeplitz structure on $K_{Z,Z}$ for $g(m) = m \log m$, allowing our RJ-MCMC procedure to train on massive datasets.

## 5 Experiments

We conduct four experiments in total. In order to motivate our model for kernel learning in later experiments, we first demonstrate the ability of a Lévy process to recover—through direct regression—an observed noise-contaminated spectrum that is characteristic of sharply peaked naturally occurring spectra. In the second experiment we demonstrate the robustness of our RJ-MCMC sampler by automatically recovering the generative frequencies of a known kernel, even in presence of significant noise contamination and poor initializations. In the third experiment we demonstrate the ability of our method to infer the spectrum of airline passenger data, to perform long-range extrapolations on real data, and to demonstrate the utility of accounting for uncertainty in the kernel. In the final experiment we demonstrate the scalability of our method through training the model on a 100,000 data point sound waveform. Code is available at `https://github.com/pjang23/levy-spectral-kernel-learning`.

### 5.1 Explicit Spectrum Modelling

We begin by applying a Lévy process directly for function modelling (known as LARK regression), with inference as described in Wolpert et al. (2011), and Laplacian basis functions. We choose an out of class test function proposed by Donoho & Johnstone (1993) that is standard in wavelet literature. The spatially inhomogeneous function is defined to represent spectral densities that arise in scientific and engineering applications. Gaussian i.i.d. noise is added to give a signal-to-noise ratio of 7, to be consistent with previous studies of the test function Wolpert et al. (2011).

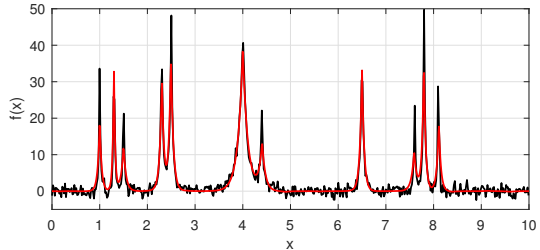

Figure 3: Lévy process regression on a noisy test function (black). The fit (red) captures the locations and scales of each spike while ignoring noise, but falls slightly short at its modes since the black spikes are parameterized as $(1 + |x|)^{-4}$ rather than Laplacian.

The noisy test function and LARK regression fit are shown in Figure 3. The synthetic spectrum is well characterized by the Lévy process, with no "false positive" basis function terms fitting the noise owing to the strong regularization properties of the Lévy prior. By contrast, GP regression with an RBF kernel learns a length scale of 0.07 through maximum marginal likelihood training: the Gaussian process posterior can fit the sharp peaks in the test function only if it also overfits to the additive noise.

The point of this experiment is to show that the Lévy process with Laplacian basis functions forms a natural prior over spectral densities. In other words, samples from this prior will typically look like the types of spectra that occur in practice. Thus, this process will have a powerful inductive bias when used for kernel learning, which we explore in the next experiments.

## 5.2 Ground Truth Recovery

We next demonstrate the ability of our method to recover the generative frequencies of a known kernel and its robustness to noise and poor initializations. Data are generated from a GP with a kernel having two spectral Laplacian peaks, and partitioned into training and testing sets containing 256 points each. Moreover, the training data are contaminated with i.i.d. Gaussian noise (signal-to-noise ratio of 85%).

Based on these observed training data (depicted as black dots in Figure 4, right), we estimate the kernel of the Gaussian process by inferring its spectral density (Figure 4, left) using 1000 RJ-MCMC iterations. The empirical spectrum initialization described in section 4.1 results in the discovery of the two generative frequencies. Critically, we can also recover these salient frequencies *even with a very poor initialization*, as shown in Figure 4 (left).

For comparison, we also train a Gaussian SM kernel, initializing based on the empirical spectrum. The resulting kernel spectrum (Figure 4, magenta curve) does recover the salient frequencies, though with less confidence and higher overhead than even a poor initialization and spectral kernel learning with RJ-MCMC.

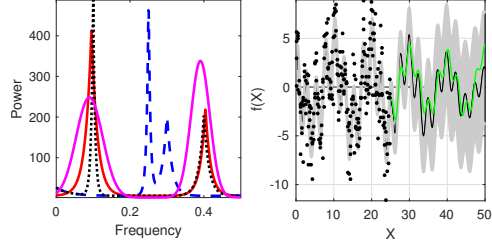

Figure 4: Ground truth recovery of known frequency components. (left) The spectrum of the Gaussian process that was used to generate the noisy training data is shown in black. From these noisy data and the erroneous spectral initialization shown in dashed blue, the maximum a posteriori estimate of the spectral density (over 1000 RJ-MCMC steps) is shown in red. A SM kernel also identifies the salient frequencies, but with broader support, shown in magenta. (right) Noisy training data are shown with a scatterplot, with withheld testing data shown in green. The learned posterior predictive distribution (mean in black, with 95% credible set in grey) captures the test data.

## 5.3 Spectral Kernel Learning for Long-Range Extrapolation

We next demonstrate the ability of our method to perform long-range extrapolation on real data. Figure 5 shows a time series of monthly airline passenger data from 1949 to 1961 (Hyndman, 2005). The data show a long-term rising trend as well as a short term seasonal waveform, and an absence of white noise artifacts.

As with Wilson & Adams (2013b), the first 96 monthly data points are used to train the model and the last 48 months (4 years) are withheld as testing data, indicated in green. With an initialization from the empirical spectrum and 2500 RJ-MCMC steps, the model is able to automatically learn the necessary frequencies and the shape of the spectral density to capture both the rising trend and the seasonal waveform, allowing for accurate long-range extrapolations without pre-specifying the number of model components in advance.

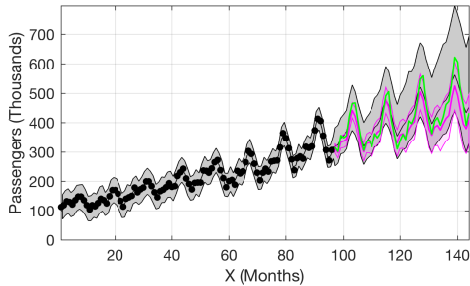

Figure 5: Learning of Airline passenger data. Training data is scatter plotted, with withheld testing data shown in green. The learned posterior distribution with the proposed approach (mean in black, with 95% credible set in grey) captures the periodicity and the rising trend in the test data. The analogous 95% interval using a GP with a SM kernel is illustrated in magenta.

This experiment also demonstrates the impact of accounting for uncertainty in the kernel, as the withheld data often appears near or crosses the upper bound of the 95% predictive bands of the SM fit, whereas our model yields wider and more conservative predictive bands that wholly capture the test data. As the SM extrapolations are highly sensitive to the choice of parameter values, fixing the parameters of the kernel will yield overconfident predictions. The Lévy process prior allows us to account for a range of possible kernel parameters so we can achieve a more realistically broad coverage of possible extrapolations.

Note that the Lévy process over spectral densities induces a prior over kernel functions. Figure 6 shows a side-by-side comparison of covariance function draws from the prior and posterior distributions over kernels. We see that sample covariance functions from the prior vary quite significantly, but are concentrated in the posterior, with movement towards the empirical covariance function.

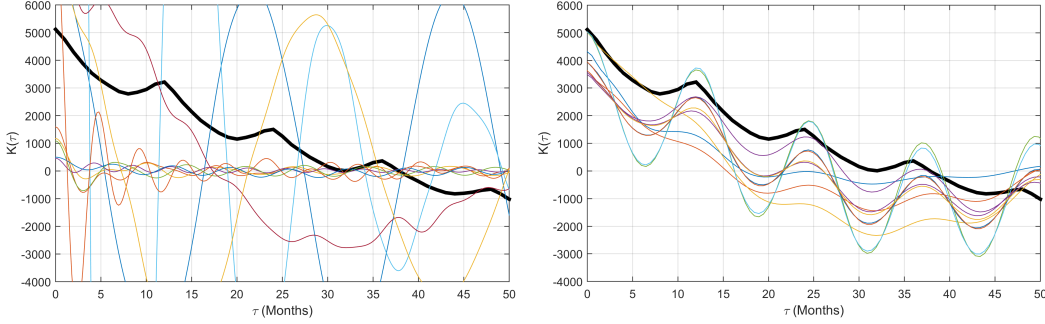

Figure 6: Covariance function draws from the kernel prior (left) and posterior (right) distributions, with the empirical covariance function shown in black. After RJ-MCMC, the covariance distribution centers upon the correct frequencies and order of magnitude.

## 5.4 Scalability Demonstration

A flexible and fully Bayesian approach to kernel learning can come with some additional computational overhead. Here we demonstrate the scalability that is achieved through the integration of SKI (Wilson & Nickisch, 2015) with our Lévy process model.

We consider a 100,000 data point waveform, taken from the field of natural sound modelling (Turner, 2010). A Lévy kernel process is trained on a sound texture sample of howling wind with the middle 10% removed. Training involved initialization from the signal empirical covariance and 500 RJ-MCMC samples, and took less than one hour using an Intel i7 3.4 GHz CPU and 8 GB of memory. Four distinct mixture components in the model were automatically identified through the RJ-MCMC procedure. The learned kernel is then used for GP infilling with 900 training points, taken by down-sampling the training data, which is then applied to the original 44,100 Hz natural sound file for infilling.

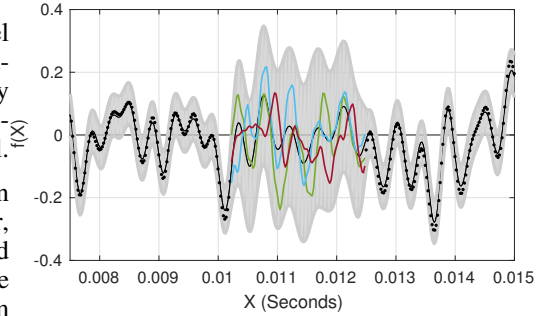

Figure 7: Learning of a natural sound texture. A close-up of the training interval is displayed with the true waveform data scatter plotted. The learned posterior distribution (mean in black, with 95% credible set in grey) retains the periodicity of the signal within the corrupted interval. Three samples are drawn from the posterior distribution.

The GP posterior distribution over the region of interest is shown in Figure 7, along with sample realizations, which appear to capture the qualitative behavior of the waveform. This experiment demonstrates the applicability of our proposed kernel learning method to large datasets, and shows promise for extensions to higher dimensional data.

## 6 Discussion

We introduced a distribution over covariance kernel functions that is well suited for modelling quasi-periodic data. We have shown how to place a Lévy process prior over the spectral density of a stationary kernel. The resulting hierarchical model allows the incorporation of kernel uncertainty into the predictive distribution. Through the spectral regularization properties of Lévy process priors, we found that our trans-dimensional sampling procedure is suitable for automatically performing inference over model order, and is robust over initialization strategies. Finally, we incorporated structured kernel interpolation into our training and inference procedures for linear time scalability, enabling experiments on large datasets. The key advances over conventional spectral mixture kernels are in being able to interpretably and automatically discover the number of mixture components, and in representing uncertainty over the kernel. Here, we considered one dimensional inputs and stationary processes to most clearly elucidate the key properties of Lévy kernel processes. However, one could generalize this process to multidimensional non-stationary kernel learning by jointly inferring properties of transformations over inputs alongside the kernel hyperparameters. Alternatively, one could consider neural networks as basis functions in the Lévy process, inferring distributions over the parameters of the network and the numbers of basis functions as a step towards automating neural network architecture construction.

**Acknowledgements.** This work is supported in part by the Natural Sciences and Engineering Research Council of Canada (PGS-D 502888) and the National Science Foundation DGE 1144153 and IIS-1563887 awards.

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
