[Supplementary Material]

# Scalable Lévy Process Priors for Spectral Kernel Learning - Supplementary Material

**Phillip A. Jang**  **Andrew E. Loeb**  **Matthew B. Davidow**  **Andrew Gordon Wilson**
Cornell University

## 1 Sampling Levy Process Priors

The following formulas in this section are taken from Wolpert et al. (2011) for reference.

Suppose the hyperparameters $\theta$ of the prior distributions for $J, \beta, \omega$, are drawn from a hyperprior distribution, $\pi_\theta(d\theta)$. Then in order to sample the Lévy prior, the follow steps are taken:

$$\theta \sim \pi_\theta(d\theta)$$
$$J|\theta \sim \text{Po}(\nu_\varepsilon^+), \qquad \nu_\varepsilon^+ \equiv \nu_\varepsilon(\mathbb{R} \times \Omega)$$
$$\{(\beta_j, \omega_j)\}_{j=1}^J | J, \theta \overset{\text{i.i.d.}}{\sim} \pi_\beta(\beta_j)d\beta_j \pi_\omega(d\omega_j)$$

The formulas for $\nu_\varepsilon^+$ and $\pi_\beta$ are determined by the specific choice of Lévy process and are given below. For computational purposes, the $\beta_j$'s are truncated at $|\beta_j \eta| > \varepsilon$ for a Poisson approximation to the true Lévy process, and $\mathbb{E}|\mathcal{L}[\phi] - \mathcal{L}_\varepsilon[\phi]|^2$ represents the $L^2$ error of the approximation for a given basis function $\phi$. Below, $E_1(z) = \int_z^\infty t^{-1}e^{-t}dt$.

### 1.1 Gamma Process

$$J \sim \text{Po}(\nu_\varepsilon^+), \qquad \nu_\varepsilon^+ = \gamma|\Omega|E_1(\varepsilon)$$

$$\beta_j \overset{\text{i.i.d.}}{\sim} \pi_\beta(\beta_j)d\beta_j, \qquad \pi_\beta(\beta_j) = \frac{\beta_j^{-1}e^{-\beta_j \eta}}{E_1(\varepsilon)}\mathbf{1}_{\{\beta_j \eta > \varepsilon\}}$$

$$\mathbb{E}|\mathcal{L}[\phi] - \mathcal{L}_\varepsilon[\phi]|^2 = \gamma\eta^{-2}\|\phi\|_2^2[1 - (1+\varepsilon)e^{-\varepsilon}]$$

### 1.2 Symmetric Gamma Process

$$J \sim \text{Po}(\nu_\varepsilon^+), \qquad \nu_\varepsilon^+ = 2\gamma|\Omega|E_1(\varepsilon)$$

$$\beta_j \overset{\text{i.i.d.}}{\sim} \pi_\beta(\beta_j)d\beta_j, \qquad \pi_\beta(\beta_j) = \frac{|\beta_j|^{-1}e^{-|\beta_j|\eta}}{2E_1(\varepsilon)}\mathbf{1}_{\{|\beta_j \eta| > \varepsilon\}}$$

$$\mathbb{E}|\mathcal{L}[\phi] - \mathcal{L}_\varepsilon[\phi]|^2 = 2\gamma\eta^{-2}\|\phi\|_2^2[1 - (1+\varepsilon)e^{-\varepsilon}]$$

### 1.3 Symmetric $\alpha$-Stable Process

$$J \sim \text{Po}(\nu_\varepsilon^+), \qquad \nu_\varepsilon^+ = \gamma|\Omega|\frac{2}{\pi}\Gamma(\alpha)\sin\left(\frac{\pi\alpha}{2}\right)\varepsilon^{-\alpha}$$

$$\beta_j \overset{\text{i.i.d.}}{\sim} \pi_\beta(\beta_j)d\beta_j, \qquad \pi_\beta(\beta_j) = \frac{\alpha\varepsilon^\alpha}{2\eta^\alpha}|\beta_j|^{-\alpha-1}\mathbf{1}_{\{|\beta_j \eta| > \varepsilon\}}$$

$$\mathbb{E}|\mathcal{L}[\phi] - \mathcal{L}_\varepsilon[\phi]|^2 = 2\gamma\eta^{-2}\|\phi\|_2^2\left[\frac{\Gamma(\alpha+1)}{\pi(2-\alpha)}\sin\left(\frac{\pi\alpha}{2}\right)\varepsilon^{2-\alpha}\right]$$

## 2 Sparsity Inducing Properties of Lévy Priors

Figure 1 illustrates the contours of the joint distribution for two independent draws of $\beta$ under different priors $\pi_\beta(d\beta)$. The contours for the gamma process would be taken from the upper-right quadrant of those for the symmetric gamma process.

Gaussian and Laplace priors on $\beta$ result in $\ell_2$ and $\ell_1$ regularization respectively. The Lévy processes in contrast yield inward curving contours, leading to a sparsity inducing effect similar to $\ell_p$ regularization with $p < 1$. Intuitively, this discourages simultaneous large values of $\beta$ more strongly than $\ell_1$ regularization unless the added basis functions significantly improve the fit.

Figure 1: Contour plot of the joint probability density function of two $\beta$ draws under different priors.

# 3 Initialization and Hyperparameter Tuning

Initialization and hyperparameter tuning can be automated by fitting the empirical spectrum of the data. It is done in the following steps:

1. If needed, de-mean the training data by subtracting a deterministic mean function such as the sample mean or best fit line. Doing so will eliminate large peaks at the origin which dominate the rest of the spectrum. The de-meaned training data $\{y_j\}_{j=1}^n$ will be the input for RJ-MCMC.

2. Compute the empirical spectral density $S_{emp}(s) = \frac{2}{n} \left| \sum_{j=1}^n y_j e^{-2\pi i s(j-1)} \right|^2$, $s \in [0, 0.5]$.
   In MATLAB, this is calculated as the first $\lfloor \frac{n}{2} \rfloor$ entries from 2*abs(fft(y)).^2/n;

3. Sample the empirical spectral density and fit a Gaussian mixture with $J_0$ components to the sampled data. A good initial guess for $J_0$ can be done by examining the number of peaks in the empirical spectrum.

$$S_{\text{Gaussian}}(s) = \sum_{j=1}^{J_0} \alpha_j \frac{1}{\sqrt{2\pi\sigma_j^2}} e^{-\frac{(s-\chi_j)^2}{2\sigma_j^2}}$$

4. Keep the frequencies $\chi_j$ from the Gaussian fit, and using least squares, fit a Laplacian basis function to each individual Gaussian component. For each $j$, one could minimize the following objective over a sample grid of points $s_k$ in $[-3\sigma_j, 3\sigma_j]$

$$\min_{\lambda_j, \beta_j} \sum_{s_k} \left[ \beta_j \frac{\lambda_j}{2} e^{-\lambda_j |s_k|} - \alpha_j \frac{1}{\sqrt{2\pi\sigma_j^2}} e^{-\frac{s_k^2}{2\sigma_j^2}} \right]^2$$

5. Form the initial spectrum with the fitted parameters

$$S_{\text{initial}}(s) = \sum_{j=1}^{J_0} \beta_j \frac{\lambda_j}{2} e^{-\lambda_j |s-\chi_j|}$$

   Plots of this initial spectrum for the airline data are shown in Figures 2 and 3.

6. Tune the hyperparameters:

   - $\lambda$ is modelled with a hyperprior Gamma$(a_\lambda, b_\lambda)$, so $a_\lambda$ and $b_\lambda$ can be estimated by maximum likelihood on the $\lambda$ parameters of the initial spectrum.
   - $\eta^{-1} \sim$ Gamma$(a_\eta, b_\eta)$ controls the expected value of coefficients $\beta_j$. For basis functions which integrate to 1, the sum of $\beta_j$'s is equal to the total area underneath the spectrum, which by Parseval's identity represents total variance of the data. Hence the sample variance of the training data can be used as an upper bound on coefficient values, and $a_\eta$ and $b_\eta$ can be set accordingly.
   - $\gamma \sim$ Gamma$(a_\gamma, b_\gamma)$ is proportional to the expected number of basis functions as shown in Section 1 and controls the sparsity of the expansions. $a_\gamma$ and $b_\gamma$ can be set to cover a range of values which encourage sparsity.
   - For the symmetric $\alpha$-stable process, $0 < \alpha < 2$ controls the heaviness of the tails in the distribution for $\beta_j$ with smaller values of $\alpha$ yielding heavier tails. $\alpha$ can be set by maximum likelihood on the initial $\beta_j$'s.
   - $\varepsilon$ can be set based on $L^2$ truncation errors as described in Section 1.

Figure 2: Initial spectrum fit after subtracting the sample mean from training data

Figure 3: Initial spectrum fit after subtracting linear trend from training data