[Reviews · NeurIPS 2017]

Reviewer 1



The paper proposes a spectral mixture of laplacian kernel with a levy process prior on the spectral components. This extends on the SM kernel by Wilson, which is a mixture of gaussians with no prior on spectral components. A RJ-MCMC is proposed that can model the number of components and represent the spectral posterior. A large-scale approximation is also implemented (SKI). The idea of Levy prior on the spectral components is very interesting one, but the paper doesn't make it clear what are the benefits with respect to kernel learning. Lines 128-135 analyse the levy regularisation behavior briefly, but this should be made more explicit and concrete. The final optimisation target doesn't seem to be explicit. A comparison to simple gaussian/laplace mixture without any priors would clarify the benefits of placing the levy prior. One would assume that the levy prior should distribute the spectral components evenly by having some 'clumped' clusters and few that are far away from each other, leading to potentially simpler models. The paper should comment more about this. There could also be some interesting connections to determinental point processes, that could also regularise the frequency components. Finally its a bit unclear what kind of Levy process the authors suggest. There are lots of options, is there some recommendation that works especially well for kernel learning? The levy kernels are sampled, but there is no experiments showing posterior distributions of e.g. J or the kernels. How much variance is there? What's the benefit of sampling, wouldn't optimisation suffice? How much slower is the method compared to e.g. the SM kernel? The experiments highlight a cases where the LKP works nicely. The first two examples actually seem to show some problems with the LKP. In fig3 the fits fall short, is this a fundamental problem with LKP or inference artifact? In fig4 the left component fit seems to be too wide. The whole 5.2. is extremely unfair experiment since the data was generated from laplacian model in the first place. Simulated data should not come from the same model. The main problem is that the SM kernel would work in all of these cases equally well or only slightly worse. There is no clear example where SM kernel isn't already good enough. In the airline example there is very little difference, and the SM kernel surely would work in 5.4. also with the SKI implementation. There's also no running time information apart from brief mention in line 335. Was 5.4. really learned with all 100,000 datapoints? Learning curves should be added. Since the main contribution of the paper is the Levy regularisation introduced to the spectral domain, a key experiment should be to compare a 'naive' L1 mixture spectral kernel to the LKP variant, and show how the LKP variant behaver better in terms of model complexity, regularisation, inference, running time or accuracy. Also comparisons to SM kernel should be included in all experiments. Nevertheless I liked the paper. The contribution is significant and well defined. The key problem in spectral kernel learning is the spectral regularisation, and this paper is one of the first papers that adresses it. Minor comments: 79: what is R^+ eq2: what is \mu_X eq9: the LKP definition is too informal

Reviewer 2



The paper is very well written. It presents an interesting idea to generalize the Gaussian process with a fixed pre-specified kernel to one which includes a distibution over kernel functions. This problem has received quite a lot of attention in recent years. As the current paper models a spectral density it most closely resembles earlier work of Wilson et al. which has received quite a number of citations indicating there is interest in these kind of methods. I also liked reading the presented method works well with simple initializations as this was a problem in earlier work.

Reviewer 3



This paper models the spectrum of a Gaussian process kernel with a location-scale mixture of Lalpacians, using the LARX method of [1] for a non-parametric prior over the coefficients. It is claimed that the levy prior promotes sparse solutions. A reverse jump mcmc method is proposed, following [1]. Inference is accelerated using a SKI approximation from [2], and results are presented on the Airline dataset and also a dataset from natural sounds. The paper is nicely written and the idea is interesting and novel, but the evaluation of the method is extremely weak. In particular, there are no quantitive comparisons, nor any mention of other kernel learning approaches (e.g. compositional kernels [3]) or other spectral approaches to GP regression (e.g. sparse spectrum GPs [5]) The first experiment presented is actually not the approach of the paper, but instead the method of [1]. It is unclear how this provides evidence for the claims of the novel contributions of the paper. The second experiment shows (qualitatively) that the model can fit a sample from its prior better than a different model. It is hardly surprising that the method of [5] fares less well on this task as Gaussian model over the spectrum is misspecified. The third experiment provides a comparison with the method of [5], but from the graph it is extremely unclear which of the two method is better, and no results on held-out data are presented. The final experiment serves only as a demonstration of scalability as the original data from the in-filled section is not shown, and the only evaluation given is that ‘the reconstucted signal has no discernable artifacts when played back as sound’. Presumably the sound file will be included in the final version, but even so, it is not easy to assess the significance of such evidence. Do other more simple approaches achieve this property? (Incidentally, the quoted sentence appears not to have been spell-checked - fortunately it seems to be the only one!) Beyond the experiments, I feel the paper should be improved by distinguishing its key contributions. It appears that there are two: the use of laplacian basis functions to model the fourier spectrum (as opposed to Gaussian in [5]), and the use of the levy prior. Is maximum likelihood over the laplacian features as effective as [5]? It should be made clear whether this is a modelling implication or chosen for convenience. Secondly, it should be explained in what circumstances a ML approach leads to overfitting, requiring the use of the levy prior. Finally, I feel that the use of ‘scalable’ in the title is a little misleading. The kernel interpolation approach is an approximation (the quality of which is not discussed), and does not extend readily to high dimensions. In summary, to improve this paper I suggest the results section is made more quantitive and emphasis on scalability is dropped and replaced by a focus on exactly why such priors are useful over a simpler ML approach. [1] Wolpert, R.L., Clyde, M.A., and Tu, C. Stochastic expansions using continuous dictionaries: Levy adaptive regression kernels. The Annals of Statistics, 39(4):1916–1962, 2011. [2] Wilson, Andrew Gordon and Nickisch, Hannes. Kernel interpolation for scalable structured Gaussian processes (KISS-GP). International Conference on Machine Learning (ICML), 2015. [3] Duvenaud, David, et al. "Structure discovery in nonparametric regression through compositional kernel search." arXiv preprint arXiv:1302.4922 (2013). [4] Quinonero-Candela, Joaquin, Carl Edward Rasmussen, and Annbal R. Figueiras-Vidal. "Sparse spectrum Gaussian process regression." Journal of Machine Learning Research 11.Jun (2010): 1865-1881. [5] Wilson, Andrew, and Ryan Adams. "Gaussian process kernels for pattern discovery and extrapolation." Proceedings of the 30th International Conference on Machine Learning (ICML-13). 2013. APA